# Current Status of Next-Generation Sequencing-Based Cancer Genome Profiling Tests in Japan and Prospects for Liquid Biopsy

**DOI:** 10.3390/life11080796

**Published:** 2021-08-06

**Authors:** Yumi Yoshii, Shunsuke Okazaki, Masayuki Takeda

**Affiliations:** Department of Cancer Genomics and Medical Oncology, Nara Medical University, Kashihara 634-8521, Japan; yoshii.yumi@naramed-u.ac.jp (Y.Y.); sokazaki@naramed-u.ac.jp (S.O.)

**Keywords:** next-generation sequencing, clinical sequencing, solid cancer, precision medicine, liquid biopsy

## Abstract

Next-generation sequencing-based comprehensive genome profiling (CGP) testing, OncoGuide NCC Oncopanel System, and FoundationOne CDx Cancer Genomic Profile have been covered by the Japanese national health insurance system since June 2019. Because CGP was initially developed to enroll patients into an early-phase clinical trial for solid tumors, its approved indications have been limited to patients who have completed the standard chemotherapy treatment. Approximately 14,000 cases have been registered with the Center for Cancer Genomics and Advanced Therapeutics as of March 2021. Measuring the drug access rate is not enough due to patients’ deteriorating condition during CGP analysis and due to the limited number of ongoing clinical trials available, although tumor-agnostic therapies, such as the use of pembrolizumab in high microsatellite-instable solid tumors and in conditions with a high tumor mutational burden (≥10 mut/Mb) as well as the use of entrectinib and larotrectinib in *NTRK* fusion-positive tumors have been approved in Japan. Moreover, since this analysis is performed using DNA derived from tumor tissue, it is difficult to perform CGP in cases in which an insufficient amount of tissue exists. Thus, noninvasive blood-based assays have been developed, and CGP panels using circulating tumor DNA from blood were approved in March 2021. However, cost, timing, and the number of tests allowed by the health system have not yet been determined. Therefore, in this review, we outline the current status and issues of CGP testing using tumor tissues as well as the expectations and limitations of liquid biopsy for use in Japanese clinical practice.

## 1. Introduction

Recent advances in the field of cancer biology have revealed that many druggable molecular alterations are shared across cancer types, and the initiation of high-throughput, next-generation sequencing (NGS) technologies has provided new opportunities to extensively analyze cancer-associated molecular alterations. In January 2015, U.S. President Barack Obama announced funding for precision medicine to promote personalized medicine. In December 2015, an “acceleration plan for cancer control” was formulated in Japan for prevention, treatment and research, and coexistence with cancer, and cancer precision medicine using NGS was designated as a key treatment and research area.

Several academic institutions and hospitals in Japan have launched NGS-based clinical sequencing research and have reported its clinical usefulness [1,2]. The clinical utility of NGS-based panel testing for lung cancer and other cancer types has been shown [3,4]. Furthermore, successful detection of a rare driver gene and the introduction of targeted therapies have also been reported [5].

Results from comprehensive genome profiling (CGP) can provide information about genomic alterations to guide the use of approved targeted therapies and ongoing clinical trials. Patients receiving biomarker-based targeted therapy had favorable treatment responses and survival outcomes with actionable alterations [6,7]. However, a randomized phase II trial comparing molecular-targeted therapy based on tumor molecular profiling versus conventional therapy in patients with refractory cancer (SHIVA trial) did not demonstrate the superiority of molecular-targeted therapy over the control treatment [8]. Therefore, it is not clear whether CGP testing for all solid cancers leads to an improved prognosis at this time.

Recent National Comprehensive Cancer Network (NCCN) guidelines offer an NGS-based CGP assay for a wide range of cancers, such as breast cancer, colon cancer, and pancreatic cancer, given that tumor-agnostic therapies, such as the use of pembrolizumab in high microsatellite-instable solid tumors and in conditions with a high tumor mutational burden (≥10 mut/Mb) as well as the use of entrectinib and larotrectinib in *NTRK* fusion-positive tumors. With the widespread approval of tumor-agnostic agents, more patients with solid tumors will have the opportunity to use NGS.

Accordingly, additional concerns exist that in some cases, it is difficult to obtain sufficient tumor tissue due to anatomical complexity, such as retroperitoneal space or pancreatic lesions. Therefore, noninvasive blood-based assays have been developed, and a CGP panel using circulating tumor DNA (ctDNA) from blood was approved in Japan in March 2021, although it is yet to be determined whether there will be regulations on the timing of use and their total costs at the timing of submission. In this review, we briefly introduce approved tissue- or liquid-based CGP testing in Japan and discuss issues related to the integration of NGS into clinical practice for patients with solid tumors.

## 2. Tissue-Based CGP Panel

In December 2018, the Pharmaceuticals and Medical Devices Agency of Japan approved the OncoGuide NCC Oncopanel System as a CGP test and the FoundationOne CDx Cancer Genomic Profile as a CGP and broad companion diagnostic for all solid tumors (Table 1) [1]. These panels became available under the universal health system in June 2019. Accordingly, we really are moving away from the one drug, one biomarker, one companion diagnostic strategy.

To consolidate Japanese precision oncology, the Ministry of Health, Labor, and Welfare (MHLW) has designated approximately 230 hospitals for cancer genomic medicine, stratifying them into three layers, i.e., designated core hospitals, designated hospitals, and cooperative hospitals. Each hospital reimbursed JPY 560,000 for administering CGP testing to each patient with an advanced solid tumor. In detail, JPY 80,000 is reimbursed when the tumor sample is submitted for CGP analysis, and the remaining JPY 480,000 is reimbursed when the test results are provided to the patient after being annotated by a molecular tumor board (MTB) at each designated core hospital or designated hospital. MTB consists of multidisciplinary specialists, such as medical geneticists, pathologists, medical oncologists, genetic counselors, and genome researchers.

Each result of CGP panel testing and patient’s information such as tumor type, chemotherapy regimen, chemotherapy-associated adverse events, and their efficacy, is required to be registered with the Center for Cancer Genomics and Advanced Therapeutics (C-CAT), a division of the National Cancer Center, to establish a nationwide cancer genome database system. Since the approved indications of CGP panels are limited to patients with solid tumors who have completed the standard treatment, if these panels were used for companion diagnostics at diagnosis, only a portion of the total CGP cost is reimbursed by insurance. Due to these panels’ complex clinical procedures and the limited number of hospitals designated by MHLW, only approximately 14,000 cases have been registered as of March 2021 in Japan.

Regarding the exit strategy, the number of clinical trials in Japan is much less than that in the United States, given that the National Cancer Institute’s Molecular Analysis for Therapy Choice precision medicine trial (NCI-MATCH) is ongoing. In this trial, patients with advanced solid tumors and lymphomas are assigned to different treatments on the basis of the molecular profile of their disease. A recent survey report from 11 designated core hospitals in Japan has shown that only 3.7% of 747 cases received genomically matched treatment [9], which is lower than that of NCI-MATCH and SHIVA trials. There are some possible reasons for this. First, the number of available clinical trials in Japan is less than that in the US and Europe. Second, there are a small number of institutions conducting clinical trials in Japan. Third, the restriction that only patients who finished all the standard care can be covered by universal health insurance makes it difficult for patients with druggable alteration to participate in clinical trials at an appropriate time. In Japan, precision medicine has been implemented in three layers as mentioned above, and clinical trials are conducted mainly at designated core hospitals. Moreover, even if druggable gene alterations are identified in patients at their local hospital, the distance from their local hospital to a designated core hospital can be a barrier for enrollment in a clinical trial, suggesting that the percentage of drug access will be lower in patients with access to only local cooperative hospitals.

To improve the drug access rate, a patient-requested therapy system (PTRS) was initiated in October 2019. A clinical research program titled “the prospective trial of patient-proposed healthcare services with multiple targeted agents based on the result of gene profiling by multigene panel test (BELIEVE study)” is led by the National Cancer Center Hospital, which examines new therapeutic opportunities for patients with genetic alterations identified with CGP tests who may have no treatment options nearby, including approved drugs or clinical trials, by utilizing PRTS. Moreover, since the indication of CGP is limited to patients who have completed the standard chemotherapy, an advanced medical care plan B (Senshin–Iryo B) is currently underway to investigate the usefulness of CGP at the time of diagnosis.

## 3. Liquid-Based CGP Panel

ctDNA has been developed as a potential noninvasive means to establish a diagnosis. Standard tissue-based analysis involves the process of assessing tumor volume and a thin section of Formalin-Fixed Paraffin-Embedded (FFPE) block; however, ctDNA can omit these processes, which can reduce turnaround time (TAT). A recent study has shown that the median TAT for liquid NGS was 11 days, whereas the median TAT for standard tissue testing was 33 days [10].

PCR-based ctDNA assays for *EGFR* mutation in non-small cell lung cancer (NSCLC) and for *RAS* mutations in colorectal cancer have been approved and implemented in clinical practice in Japan. The FoundationOne Liquid CDx Cancer Genome Profile, Japan’s first CGP for solid tumors using blood samples, was approved in March 2021 (Table 1). It is yet to be determined whether there will be regulations on the timing of use, such as after the completion of standard chemotherapy, and their total costs. Since tumors are a heterogeneous population (i.e., primary tumors and metastatic sites have different genomic profiles) [11], plasma may capture the genomic heterogeneity of multiple, spatially separated tumor clones.

Not all patients with advanced cancer can access tissue-based CGP due to limited tumor-derived DNA and RNA. Even in cases where sufficient tumor tissue is not available, such as in pediatric cancer, retroperitoneal tumors, pancreatic cancer, and cholangiocarcinoma, a liquid CGP panel can capture the ctDNA and RNA. In addition, resistance mechanisms have been widely investigated in driver-positive cancers. For example, clinical trials of tepotinib for MET-positive *EGFR* mutation-positive lung cancer after resistance to EGFR-TKIs are underway. In addition, re-biopsy of a metastatic lesion is not always feasible in clinical practice and can result in unpleasant patient complications due to its invasive nature. In such cases, noninvasive liquid biopsy can be a potent option for treatment. It has been reported that the recurrence rate after surgery is higher when mutations are detected in ctDNA after surgery [12]. For example, the presence of minimal residual disease (MRD) may be a factor for determining the need for more intensive anticancer drug treatment after surgery.

However, it is also necessary to fully understand the weaknesses of liquid biopsies. In this process, it is assumed that the tumor is sufficiently flowing in the bloodstream. A recent systematic review on the comparison between liquid-based and tissue-based biopsies using targeted NGS in advanced NSCLC patients has shown that liquid biopsies might be unable to fully substitute their tissue counterparts in detecting clinically relevant mutations due to some disconcordance of driver mutations, such as *EGFR*, *ALK*, *ROS1*, and *BRAF* mutations [13]. According to the recently published policy recommendation “Cancer genome profiling assay using circulating tumor DNA in blood” established by the Joint Task Force for the Promotion of Genome Medicine (created by the Japanese Society of Medical Oncology, the Japanese Society for Clinical Oncology, and the Japanese Cancer Association), an expert panel (MTB) is mandatory for plasma CGP assays and tissue CGP assays, which are currently reimbursed by the national health insurance system [14]. In the plasma-only analysis, it is difficult to distinguish tumor-derived ctDNA mutations from cloned hematopoietic mutations in normal cells. It may be useful for MRD, but insurance approval will probably not be available immediately after surgery.

## 4. Conclusions

Because the CGP test is performed only after the completion of standard chemotherapy in Japan, the drug access rate is low due to the limited number of ongoing clinical trials. CGP analysis is performed based on the DNA derived from tumor tissue, and it is difficult to perform the test in cases where there is an insufficient amount of tissue sample available for analysis. Thus, noninvasive, blood-based assays have been developed, including a CGP panel using ctDNA from blood. Further advances in this type of test will require the real-time acquisition of knowledge about ctDNA in clinical practice.

## Figures and Tables

**Table 1 life-11-00796-t001:** Approved NGS panels in Japan.

	FoundationOne CDx	FoundationOne Liquid CDx	NCC Oncopanel
No. of genes	324	324	124
Sample	DNA	ctDNA	DNA
Paired sample (control)	N/A	N/A	Blood
FDA approval date	Nov, 2017	Oct, 2020	N/A
PMDA approval date	Dec, 2018	Mar, 2021	Dec, 2018
Companion diagnostic	Yes	Yes	No

NGS: next-generation sequencing; ctDNA: circulating tumor DNA; FDA: U.S. Food and Drug Administration; PMDA: Pharmaceuticals and Medical Devices Agency of Japan; N/A: Not Applicable.

## Data Availability

Not applicable.

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
