# Peer review of "Current Status of Next-Generation Sequencing-Based Cancer Genome Profiling Tests in Japan and Prospects for Liquid Biopsy"

_life, 2021, doi:10.3390/life11080796_

Round 1

Reviewer 1 Report

Yoshii et al. described the current situation on cancer genome profiling test in Japan. This is a well summarized and arranged comment, but some points could be further addressed.

  1. In the Table 1, the date when FDA and/or PMDA approved each test could be added. Foundation One liquid was approved in March 2021, later than the other two tests.
  2. Line 104. Only 3.7% (28 cases) of the cases received genomically matched treatment according to the ref. 9. ERBB alterations, FGFR alterations, EGFR mutations and NTRK fusions mainly comprises these matched cases. Authors could provide these details and compare the other similar studies such as NCI-MATCH and the ref.8. Is this rate (3.7%) comparable with other basket studies? If the rate in Japan is lower than other studies, authors are warranted to describe the possible reason of these discrepancies.

Author Response

We thank the reviewer for insightful comments, which we feel have helped us to improve our manuscript. Our specific responses to the points raised are as follows:

  1. According to the reviewer’s suggestion, we have now provided the date of PMDA approval in the Table 1.

  1. As the reviewer points out, the drug access rate in the survey report from 11 designated core hospitals in Japan is lower than that of NCI-MATCH and SHIVA trial. We have now provided some possible reasons in the revised manuscript (p. 3, lines 113-118).

Reviewer 2 Report

This manuscript gives a brief introduction of clinical application status of next-generation sequencing-based comprehensive genome profiling (CGP) testing in Japan. NGS has become an important genomic testing technique in cancer medicine. This short report presents ongoing status of CGP and strategy in clinical practice in Japan. It is interesting and will attract people in this field.

The manuscript is well written and easy to read.

In lines 54 and 71, the words of comprehensive CGP may be considered to delete the word of comprehensive because CGP already indicates “comprehensive”.

Author Response

We thank the reviewer for insightful comments, which we feel have helped us to improve our manuscript. Our specific responses to the points raised are as follows:

  1. According to the reviewer’s suggestion, we have now deleted the word “comprehensive” in the revised manuscript.